# Morphological Variations of Wild Populations of *Brycon dentex* (Characidae, Teleostei) in the Guayas Hydrographic Basin (Ecuador). The Impact of Fishing Policies and Environmental Conditions

**DOI:** 10.3390/ani11071901

**Published:** 2021-06-26

**Authors:** Ana Gonzalez-Martinez, Carmen De-Pablos-Heredero, Martin González, Jorge Rodriguez, Cecilio Barba, Antón García

**Affiliations:** 1Department of Animal Production, Faculty of Veterinary Sciences, University of Cordoba, 14071 Córdoba, Spain; agmartinez@uco.es (A.G.-M.); cjbarba@uco.es (C.B.); 2Department of Business Economics, Applied Economics II and Fundamentals of Economic Analysis, ESIC Business & Marketing School, Rey Juan Carlos University, Paseo de los Artilleros s/n, 28032 Madrid, Spain; carmen.depablos@urjc.es; 3Department of Animal Production, Quevedo State Technical University, Av. Quito km, 1 1/2 vía a Santo Domingo de los Tsáchilas, Quevedo, Los Ríos 120501, Ecuador; mgonzalez@uteq.edu.ec (M.G.); jrodriguez@uteq.edu.ec (J.R.)

**Keywords:** discriminant analysis, morphological differentiation, conservation, native resources

## Abstract

**Simple Summary:**

The conservation status of a native fish species is often a key indicator of the state of habitat alteration, which supports strong anthropogenic disturbance. Ecuador contains the Guayas basin, the largest basin in the Pacific Ocean, which is a biodiversity reserve. However, there is little information regarding the morphometric characterization of *Brycon dentex* and its variations within this basin, although its plasticity has been proposed as an indicator of the maintenance of biodiversity. The goal of this study was to analyze the effects of anthropogenic activity and habitat modification on the morphological variation of *Brycon dentex* and to determine the usefulness of discriminant analysis in the morphometric differentiation of three populations of *Brycon dentex* in Ecuador. The *Brycon dentex* morphometric model could be used as a framework in conservation and, thus, an indicator of habitat status by quickly detecting changes in fish shape.

**Abstract:**

The Guayas, located in Ecuador, is the largest basin in the Pacific Ocean and has an inventory of 123 native freshwater species. Most of these are endemic species that are threatened or at-risk due to anthropogenic activity and the modification, fragmentation, and destruction of habitats. The aim of this study was to determine the morphometric variation in three wild populations of *Brycon dentex* in the Guayas basin rivers and their connections to fishing management and environmental conditions. A total of 200 mature fish were captured, and 26 morphometric parameters were measured. The fishing policies (Hypothesis 1) and environmental conditions (Hypothesis 2) were considered fixed factors and were validated by t-tests. The morphological variation among the three populations (Hypothesis 3) was validated through a discriminant analysis. Fishing policies and resource management were found to generate morphological differences associated with body development. In addition, the environmental conditions were found to influence the size and structure of *Brycon dentex* populations. The analyzed populations were discriminated by the generated morphometric models, which differentiated Cluster 1 (Quevedo and Mocache rivers) with high fishing pressure from Cluster 2 (Pintado river) with medium–low fishing pressure. Morphometric differentiation by discriminant analysis is a direct and economic methodology that can be applied as an indicator of diversity maintenance.

## 1. Introduction

The ecological theory of diversification [1] and studies of wild populations explain how changes in environmental factors could induce changes in behavior, morphology, and physiology [2,3]. Selection pressure in new environments favors the divergence of populations, and there is a strong link between environmental variations and the morphological diversification of a population [2,4,5]. Furthermore, habitat modification may result in changes in the composition, geographical spread, and population structure of a species [6,7,8], while diversity is considered to be an indicator of ecosystem restoration [9]. In this sense, fishing policies in each country seek food sovereignty and the maintenance of genetic resources and biodiversity through economic incentives and regulatory measures [10]. The construction of habitat conservation indicators in Latin America is complex due to a lack of data and the absence of characterization and productive behavior studies [11]. The characterization of animal genetic resources covers all activities associated with the identification and quantitative and qualitative descriptions of populations, as well as the natural habitat and production systems to which they are adapted [12]. In this sense, morphological analysis has been widely used for breed and population characterization [13,14]. It has recently been used in comparative morphometric studies of native freshwater species from Ecuador. It could also be useful for the implementation of a livestock development program and as a restoration indicator. The evaluation of morphological variations in a native freshwater species living under different conditions within the same habitat could help to identify the factors responsible for these differences [15]. According to Dauda et al. [16], this knowledge is key for the management of fisheries, as having stocks with different life-history traits is essential to enhance stock management programs [17]. A study by Ndiwas et al. [2] noted that the geographical spread of a species across a broad range of environmental conditions is accompanied by equally diverse morphological variations that are strongly correlated to the environmental conditions. Many factors can cause morphological variations between populations, and they are hard to identify [18]. Previous studies using geometric morphometrics to investigate variations in fish have found differences based on sex [15], diet composition [19,20], the geographic location of a population [21], and the habitat and water characteristics [3,22]. Morphometry is a cost-effective technique that is frequently used for the differentiation of populations [13,23,24] and is employed to describe fish body shapes, delineate stock status, discriminate between fish populations, and link ontogeny with functional morphology [18]. It is also necessary to collect actual biological information on fish such as data on their ecology, evolution, behavior, and stock assessment [25,26] associated with diverse species, breeds, and populations [27]. Thus, the characterization of native fish and the level of diversity are essential for the development of conservation programs and knowledge of morphometric characteristics. This constitutes the first step in the classification of animal genetic resources [16,28].

Developing countries are home to most fish species in the world, although a large proportion of fish species remain unassessed due to insufficient scientific studies. Ecuador, with 951 native freshwater species, is considered to be a biodiversity reserve [29]. The fluvial network of Ecuador is complex and diverse, and the Guayas river hydrographic basin (CHG), covering 53,299 km^2^ (Figure 1), is the largest Pacific Ocean basin in South America [30].

*Brycon dentex* Günther 1860 (pez dama), from the Bryconidae family, is a native species that is widely distributed in western Ecuador in different rivers of the CHG [31,32,33], in the Tahuín dam near Peru [34], and in the North of Perú [35]. In a previous study, our research group conducted a preliminary morphology characterization of *Brycon dentex* [36]. Morphologically, this species has a single dorsal fin and two bifurcations in the caudal fin, a total length of 51 cm, and a powerful upper jaw. It is considered an omnivorous species [33,37]. Sampling in natural environments has shown that it can reach sexual maturity at lengths of 20–26 cm [37]. Its conservation status is of “least concern” (LC), and it is included in the IUCN Red List.

The subject of our study, *Brycon dentex*, is widely spread across a broad geographical range of ecological conditions accompanied by equally diverse morphological variations in Ecuador. It is included on the IUCN Red List as LC, although there are insufficient data for characterization. *Brycon dentex* has an omnivorous mode of feeding and a strong ability to rapidly adapt to different environmental conditions. In addition to characterizing *Brycon dentex*, it is of great interest to relate its morphological variability to biodiversity maintenance. The variation among the stocks of river populations could be a consequence of phenotypic plasticity in response to unusual hydrological conditions [2,17].

Ferrito et al. [38] and Mir et al. [39] conducted similar studies on other freshwater species, and Dasgupta et al. [40] stated that morphological discrimination in various populations are strongly influenced by habitat differences. Growth variations also occur in response to different habitats [41]. Under the hypothesis that the morphological differences between populations of native freshwater species could be used as bioindicators, the causes of these differences were classified into anthropogenic and habitat modifications [42]. Knowledge in this area can be used as a tool for both the smart and ecological management of resources [43,44]. In addition, there is a lack of knowledge about the morphological characterization of *Brycon dentex* and the variation in the traits of different rivers in the Guayas basin. In this context, this study aimed to contribute to the attainment of an adequate management ecosystem equilibrium and the characterization of animal genetic resources.

Therefore, we investigated whether three wild populations of *Brycon dentex* in the CHG have undergone significant morphological diversification and whether these differences can be related to fishing management and environmental conditions. Knowledge of the phenotypic variations in relation to environmental modifications could be used to identify key factors for policy makers in terms of the development of both diversity conservation programs and sustainable fishing practices.

The general hypothesis of this study was that fishing policies, resource management, and environmental conditions influence the differentiation of populations due to the phenotypic plasticity of *Brycon dentex*. Subsequently, three hypotheses were formulated, as displayed below:

**Hypothesis** **1.**
*Fishing and resource management practices (such as fishing methods and pressure, respect of the closure periods, land use, endogeneity level, and competition with native and non-native species) influence the morphological variation of Brycon dentex in the CHG. The null hypothesis (H_0_) was that there would be an equality of means between the populations of Brycon dentex in the Pintado and Mocache rivers (H_0_: μ_1_ = μ_3_). In contrast, the alternative hypothesis was that significant differences would exist due to fishing management practices (H_1_: μ_1_ ≠ μ_3_).*


**Hypothesis** **2.**
*The physical environmental conditions influence the morphological variation of Brycon dentex. The null hypothesis (H_0_) was that there would be an equality of means between the populations of Brycon dentex in the Quevedo and Mocache rivers (H_0_: μ_2_ = μ_3_). In contrast, the alternative hypothesis was that significant differences would exist due to a change in environmental conditions (H_1_: μ_2_ ≠ μ_3_).*


**Hypothesis** **3.**
*The fishing management and environmental conditions influence the morphological variation of Brycon dentex in the CHG. The relationships among three populations were analyzed through discriminant analysis, where three groups were identified: Population 1 (Pintado river) with a medium fishing pressure and a low flow velocity, Population 2 (Quevedo river) with a high fishing pressure and white water or water containing a high concentration of oxygen, and Population 3 (Mocache river) with a high fishing pressure and a low flow velocity.*


## 2. Materials and Methods

### 2.1. Data Collection and Study Area

A stratified sample of 200 adult specimens from three populations of wild *Brycon dentex*—from the Pintado (population 1; sample size = 50), Quevedo (population 2; sample size = 93), and Mocache (population 3; sample size = 57) rivers—was chosen. The non-representative specimens of *Brycon dentex* with small size, fish of other species, mutilated, and young individuals were immediately returned to the river. These areas are in the CHG (Figure 1). The Pintado river is located in the northwest of Manabí province in the “La manga del Cura” area, and it flows into the Daule-Peripa dam. The Pintado river is born as a continuation of the Pupusá river and borders the canton of Carmen with very slow waters, high turbidity, and little oxygenation. The water characteristics of the Pintado River are shown in Table 1. These values are similar to those of the Mocache river (pH, electric conductivity, and temperature). The river delimits a protected natural area of tropical humid forest inhabited by native farmers called “montuvios”. The fishing pressure on native species is medium–low, and resource management practices favor “land sharing or wildlife-friendly agriculture”, so farmers in this area apply low-intensity, more environmentally friendly agricultural practices [45]. The Quevedo river, which originates in the foothills of the Andes mountain range in the mid–high area of the basin, contains white, fast, and highly oxygenated waters and has a length of 163 km. The Mocache river is located in the mid–low area of the Guayas delta basin and contains slow water, a low oxygen level, and a high level of dissolved solids. Both rivers (Quevedo and Mocache), located in the province of Manabí, have high fishing pressure for native species [46].

The physical attributes (pH, temperature, color, conductivity, turbidity, dissolved oxygen, and total dissolved solids), chemical indicators (chlorides, alkalinity, nitrates, and ammonium), and biological indicators (phytoplankton) of the water in the three rivers show values within the recommended ranges for freshwater life [5,7,31]. Table 1 shows the main water quality characteristics in the three rivers sampled.

The most commonly used types of fishing gear are hand-held lines with hooks (10% of catches); cloth or tape mode nets (20%) with a length of 100–150 m, a height of 4–5 m, and a mesh light diameter of ¾–1 inch; trammels (30%) with a mesh size of between 2 and 3½ inches; casts (35%) with a mesh size of between ½ and 1 inch; and others such as harpoon hooks (5%). Manual riverbank throw nets, trammel nets, and fishing spears are the most frequently used fishing methods in the Pintado river. By contrast, trawl lines, riverbank manual throw nets, trammel nets, throw nets, and fishhooks are the most commonly used methods in the Quevedo and Mocache rivers.

Balsas and canoes are ancestrally used by fishermen, while bongos (15–70 cv; measurement of power where 1 cavallo vapore (cv) = 0.98632 horsepower (hp)) boats (25–50 cv), and lanchas (30–180 cv) use motors from low to high strengths and are autonomous. Previous research by Pacheco-Bedoya [50], FAO [51], MAGAP [52], and Ochoa Ubilla et al. [53] widely analyzed the diverse traditional fishing tackle methods utilized in the CHG. Rural communities were found to use highly diverse fishing vessels. In the Pintado river, the bongo and boat were identified as the most commonly used fishing vessels. In the Quevedo and Mocache rivers, the balsa, canoe, bongo, boat, and lancha were identified as the most commonly used.

*Brycon**dentex* specimens (weight > 56 g; body length > 12.38 mm) were caught by fishermen between January and March 2019. The capture, transport, and stunning of specimens were conducted following the recommendations of Gonzalez-Martinez et al. [13]. A veterinary practitioner supervised the animals’ welfare in each research step.

### 2.2. Body Measurements

Morphometric trait data were collected using an ichthyometer with graduated digital calipers and tape with a precision of 0.01 mm. To avoid errors, the same researcher measured all fish starting from the left side, except for the widths and perimeters, following the conventional method described by Diodatti et al. [54]. A total of twenty-six morphometric measurements based on 23 landmarks and five meristic counts were obtained (Figure 2 and Table 2), in agreement with the methodology used to assess native species in Ecuador [11,55,56].

### 2.3. Fulton Condition Coefficient (K)

The Fulton condition coefficient (K) is a widely used indicator of animal welfare in fisheries and general fish biology studies to quantify variations in fish populations [57]. It was calculated with the equation K = 100 × (BW/SL^3^), where BW is the total weight (g) and SL is the standard length (cm).

### 2.4. Statistical Analysis

The morphometric characters were standardized in accordance with the work of Elliot et al. [58]. The efficiency of the transformation-adjusted size was evaluated by testing the correlation significance between each transformed variable and the standard length. The Kolmogorov–Smirnov and Bartlett tests were performed before the analyses to verify the normality and equality of the data variance (homoscedasticity). The KMO sampling adequacy test showed a value of 0.6, while the Bartlett test showed a satisfactory probability value (*p* < 0.001), thus indicating that the analysis was suitable [13,14]. The total length was obtained by calculating the arithmetic mean of both total lengths due to the bifurcated caudal fin of this species (Figure 2). Morphometric characteristics (original and adjusted) were compared by Student *t*-tests (hypotheses 1 and 2), whilst meristic counts were compared with the Kruskal–Wallis test. Sex was not considered as a fixed effect, since no significant differences between males and females have been found previously. To contrast Hypothesis 3, the DISCRIM procedure was used to perform a canonical discriminant analysis of size-adjusted geometric morphometric data using the three populations as the grouping variable. The probabilities of entering and staying in the model were both set at *p* < 0.05. The selection of the most discriminant variables was conducted by applying the F-Snedecor, Wilks’ lambda, and 1-Tolerance methods. The correct assignment percentage was considered, and the Mahalanobis distances are represented graphically as clusters. Statistica 12.0 for Windows software was used to perform the statistical analyses (StatSoft, Tulsa, OK, USA).

## 3. Results

The fish in the *Brycon dentex* population were shown to have an average body weight of 154.47 g; standard and total lengths of 15.65 and 21.46 cm, respectively; and an average head length of 5.65 cm (Table 3). The morphometric traits had a medium level of homogeneity with low coefficients of variation. The Pearson correlation coefficients between morphometric measures were high and significant (*p* < 0.05) (data not presented). The highest weight and K condition factor were obtained in the Pintado river (172.61 g and 5.10, respectively), and the lowest were obtained in the Quevedo river (137.95 and 3.53 g, respectively) (Table 3).

In relation to Hypothesis 1, the results of the morphometric comparisons between two wild populations from the Pintado and Mocache rivers with different fishing management practices are shown in Table 3 (A × C). There were no significant differences in body weight or the condition factor between the two populations (*p* > 0.05). Eleven of the 25 morphometric measures showed significant differences (*p* < 0.05) for the populations from the Pintado and Mocache rivers (Pre-PvL, Pre-AL, DFRL, AFL, UJL, AC3, P1, P3, LC1, LC2, and LC3). Regarding Hypothesis 2, Table 3 (B × C) shows the influence of environmental conditions on morphological differentiation in populations of *Brycon dentex*. Significant differences between the Quevedo and Mocache rivers were found for 10 morphometric variables: TL, SL, HL, Pre-OL, Pre-DL, Pre-PcL, Pre-PvL, Pre-AL, AFL, and UJL.

Meristic traits based on the number of dorsal, pectoral, pelvic, anal, and caudal fin rays showed mean values of 9.98, 10.41, 7.81, 29.39, and 19.01, respectively (data not presented). Coefficients of variation were low. The river did not significantly (*p* > 0.05) affect the meristic traits, with close values shown for the three populations.

The discriminant function, obtained with 24 measurements, showed that were 14 (58.33%) variables were accepted in the model and 10 measurements (41.67%) were significant (*p* < 0.05). According to the results of the F-Snedecor, Wilks’ lambda, and 1-Toler methods, there were six major discriminant variables in the model (25%): the anal fin length (AFL), body perimeter 3 (P3), body depth 3 (AC3), total length (TL), upper jaw length (UJL), and pre-pelvic fin length (Pre-PvL) (Table 4). The relationships among discriminant variables showed a specific morphology model for each group of *Brycon dentex*, with a percentage of correct assignment of 68.84. The morpho-structural differences among the three analyzed fish populations were visually obtained from morphometric measurements through a graphical representation of Mahalanobis distances (Figure 3). There was a first cluster grouping of specimens from the Quevedo and Mocache rivers, and there was a second cluster made up of specimens from the Pintado river.

## 4. Discussion

With a total average length of 21.46 cm, the *Brycon*
*dentex* individuals caught in our study were smaller than those recorded by Revelo [59] (23.7 cm) and Revelo and Laaz [33] (23.3 cm). According to Ochoa Ubilla et al. [53], the fishing gear used and the fishing pressure to which a river is subjected to determine both the frequency of capture and the size of the captured fish.

The results of this study showed significant differences between rivers; however, a low standard error and low variability within each river were found. This indicates the presence of homogeneity due to a potentially shared morphological plasticity and parallel adaptation to similar habitat types [22]. The availability and diversity of fish are indicators of the degrees of human intervention and habitat modification [60]. There are many factors that can be used to understand the morphometric variations among the three *Brycon dentex* populations, including food availability, strong competition with other non-native species (*Oreochromis* spp.), overfishing through several fishing methods (riverbank manual throw nets, trammel nets, and fishing spears), unsuitable fishing policies, land use, agricultural practices, and the destruction of Ecuador river habitats [6,8,45,61,62]. Moreover, food availability and pollution are conditioned by different abiotic parameters, including current flow, turbidity, and dissolved oxygen concentration [61]. In the Quevedo river, Prado et al. [61] identified a high level of diversity for phytoplankton and a low diversity for zooplankton. However, the Mocache and Pintado rivers presented greater variability and a greater proportion of zooplankton, insects from *Baetis* spp., and larvae from *Astyanax* spp. [61]. Furthermore, in the Mocache river, the presence of *Polymyxus coronalis* has been related to eutrophic waters.

Hypothesis 1, that fishing policies and resource management influence the morphological differentiation of *Brycon dentex* populations between the Pintado and Mocache rivers, was accepted for 11 of the 25 variables. For the traits linked to the body development of both populations (depth, perimeter, and width of body), significant differences were found. The morphological differences in BW, K, TL, and SL between the Pintado and Mocache rivers were not significant. Contrary to expectations, there were no differences in the size and structure of the fish between these rivers (Hypothesis 1), most likely because the local fishing regimes are closer than initially considered [29,47,59]. *Brycon dentex* populations in the Pintado river were found to have greater body depths, and specimens from the Mocache river had greater perimeters and widths. In the Pintado river, the fishing pressure was low, as this area acts as a reservoir of native freshwater species and encourages the use of environmentally friendly agricultural practices.

The differences between specimens in the rivers were influenced by the high fishing pressure in some areas, the overfishing of native species in the Mocache river, and the introduction of non-native species [29,46,59,63]. The role of non-natives species was not tested in this study, although it has been studied widely. According to Canonico et al. [64], the introduction of tilapia has had a large effect on native biodiversity, because tilapias are fast-growing and tolerant of a range of environmental conditions. This species readily adapts to changes in salinity levels and oxygen availability, can feed at different trophic levels, and (under certain circumstances) can tolerate overcrowding. In addition, tilapias are reproductively active for long periods—for most of the year in some places. They have short reproductive cycles and have been observed to spawn year-round in the wild with a higher frequency than most fish. They are also competitors with native species for food and space, and, therefore, native fish have fewer resources for somatic growth than at non-invaded sites. Hypothesis 2, which states that the differences in environmental conditions between the Quevedo and Mocache rivers influence the size and structure of *Brycon dentex*, was accepted in 12 out of 25 variables. Body weight and the condition factor (K), which indicates the nutritional status of fish, were higher in the Mocache river population (165.82 g and 4.93, respectively). In contrast, the total length, standard length, and eye diameter were higher in specimens in the Quevedo river. Specimens of greater size and with a greater hydrodynamic structure were found in the Quevedo river. These factors enhance their ability to survive and give them speed during flight from predators [65]. In terms of variability interpretation, there are multifactorial causes [66], such as food availability [33], seasonality [67], and the interaction between the two [68]. Therefore, the different morphologies found for populations in the Quevedo and Mocache rivers might be due to the fact that the Mocache river is slower, shallower, has greater turbidity, and has a higher level of photosynthesis and oxidative reactions [6,31]. Phytoplankton and zooplankton were identified as the most dominant foods [7]. Food availability seems to be the most relevant factor in the Mocache river, because this fish is an opportunistic feeder that switches from one diet to another according to food availability [19].

The three rivers showed differences in the morphology of *Brycon dentex* with factors associated with the fragmentation and deterioration of the ecosystem and a high fishing pressure existing in areas with the highest vulnerability level. According to Ochoa Ubilla et al. [53], the native species fishing pressure in the Quevedo and Mocache rivers is very high due to the large number of fishing cooperatives and the high fishing effort during the fishery season. The use of illegal fishing gear has also increased, thus contributing to the rivers’ deterioration [37,59]. Our findings are in agreement with those of Youson et al. [69] and Escanta-Molina and Jimenez-Prado [3], who related morphological differentiation to increases in temperature, suspended solids, and water turbidity, as well as decreases in pH, dissolved oxygen, and alkalinity. Ekaete [70] related morphology to the temperature and the amount of oxygen that fish absorb through their gills, the type of vegetation cover in the river, and the availability of food.

Hypothesis 3 was partially accepted. Though the discriminant model was accepted, Mahalanobis distances only differentiated two groups. Six morphologic variables with a high discriminant power were selected. The three analyzed *Brycon dentex* populations were discriminated by the generated morphometric models. Melvin et al. [71] and Ujjania and Kohli [43] pointed out the importance of using genetic and environmental interactions to complement the morphology approach, although it is not always easy to explain the causes of morphological differences between populations [72]. Surprisingly, the use of Mahalanobis distances only allowed us to differentiate two groups based on the morpho-structural model: Cluster 1 (Quevedo and Mocache rivers) with high fishing pressure and Cluster 2 (Pintado river) with medium–low fishing pressure.

AFL, UJL, and Pre-PvL variables were common in the differentiation of both factors. The AC3 and P3 traits were found to be associated with body development and the fishing management factor. The TL variable was linked with size, structure, and the water characteristics in each river. Native species tend to maintain a constant body mass and adaptively respond to anthropogenic and environmental conditions. *Brycon dentex* specimens in the Quevedo river (fast and highly oxygenated waters) are larger and narrower than those in the other rivers. On the contrary, fish in warm, slow, and less oxygenated waters (Mocache and Pintado rivers) respond by modifying their body development (width and deep). These results agree with those of Cavalcanti et al. [73] and Olaya Carbó et al. [74]. The CHG is made up of several fragile river ecosystems that are permanently subjected to risk factors such as the modification, fragmentation, and destruction of habitats; the introduction of non-native species (*Oreochromis* spp.); overfishing; environmental contamination (herbicides, heavy metals, etc.); the development of large-scale intensive forestry practices; a loss of modification of the natural hydrological regime, including basin river connectivity; and, finally, climate change [6,30,75,76]. Uncontrolled exploitation is a process that could lead to dangerous situations in terms of fish extinction [3,31,33,37,59]. Anthropogenic activities directly influence fishing policies, land use, and the indirect method of modifying the limnological characteristics of different ecosystems [77]. The adequate regulation of human activity, which contributes to the risk factors mentioned above, is essential for the conservation of these ecosystems [10,60]. These factors have great importance in resource conservation for the development of fisheries and sectorial policies [29,37,78].

Native freshwater populations are very sensitive to environmental changes and quickly adapt to morphological changes [79]. Therefore, the discriminant model built for *Brycon dentex* could be used as an indicator of habitat conservation and endogeneity degree [3,15,75]. The characterization carried out in this study was preliminary, since it only considered the morphological aspects and investigated variations among different habitats in terms of fish morphology. The ideal scenario would be to conduct studies with molecular markers to identify genetic relationships among populations from different habitats and specimens captured at different points, but, at the moment, little study on native species from Ecuador has been done. Morphometric methodology is a direct, simple, and low-cost method, so its in situ use is recommended in rural communities and developing countries [13]. It also could be used to assess fishing pressure and the effectiveness of sustainable policies [33].

The interactions between factors were not considered in this research, nor were the action mechanisms existing for each factor. Additionally, the characterization of *Brycon dentex* lacked a genetic analysis of the species.

## 5. Conclusions

The three populations of *Brycon dentex* in the CHG showed morphological differences due to fishing management and environmental conditions. Both factors are considered to be direct drivers of diversity maintenance.

The different fishing policies and resource management practices present in the studied rivers have generated morphological differences that are associated with body development. Under the different environmental conditions present in the rivers, differences in traits related to the size and structure of fish have primarily arisen.

The analyzed populations could be discriminated using the generated morphometric model, showing that discriminant analysis is a useful way to differentiate populations. Six morphometric measurements were found to have the greatest discriminant power and were deemed to be appropriate for population discrimination. These variables were the anal fin length, body perimeter 3, body depth 3, total length, upper jaw length, and pre-pelvic fin length.

The *Brycon dentex* morphometric model could be a useful framework for the conservation of the species, so it could indicate habitat status by quickly detecting changes in fish shape. A very small number of easily obtained morphometric traits in each population could be used to determine the biodiversity status, and this information could also be used in the implementation of fishing management policies in Ecuador’s rivers. This model could be extended for use in other rivers in Latin America.

## Figures and Tables

**Figure 1 animals-11-01901-f001:**
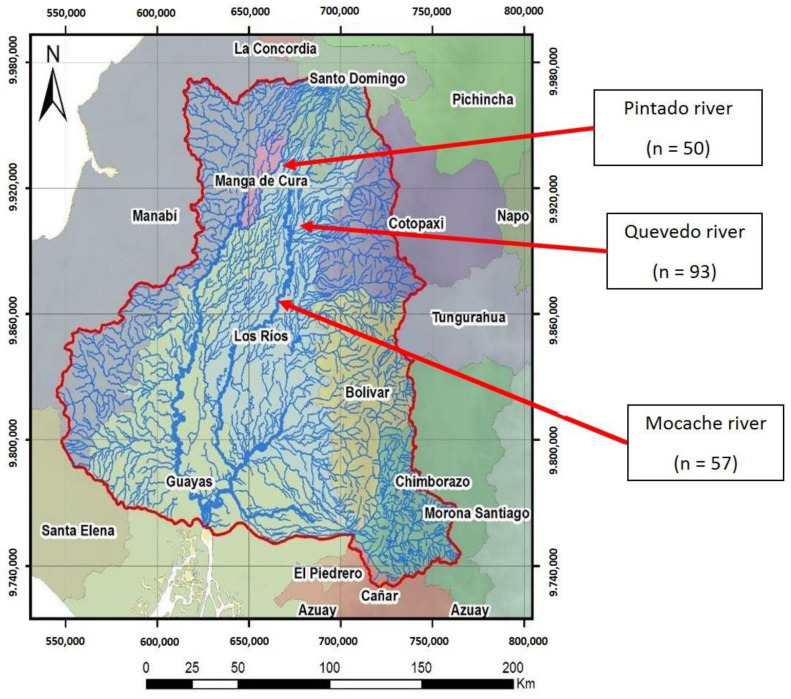
Guayas hydrographic basin (CHG). Site 1: Pintado river; Site 2: Quevedo river; Site 3: Mocache river (n = sample size).

**Figure 2 animals-11-01901-f002:**
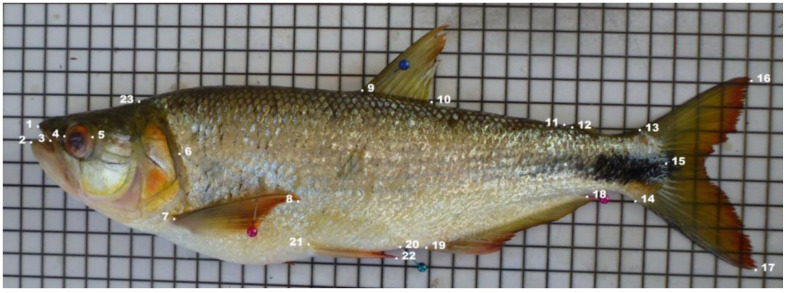
Locations of 23 anatomical landmark points showed in the left view of the *Brycon dentex*. Landmark points: 1: highest cranial point of the upper pre-maxilla; 2: highest cranial point of the lower pre-maxilla; 3: commissure of the mouth; 4: anterior edge of the eye; 5: posterior edge of the eye; 6: end of the operculum; 7: origin of the pectoral fin; 8: end of the pectoral fin radius; 9: origin of the first dorsal fin; 10: end of the dorsal fin; 11: origin of the second dorsal fin; 12: end of the second dorsal fin; 13: dorsal origin of the caudal fin; 14: ventral origin of the caudal fin; 15: highest cranial point of the caudal peduncle; 16: highest caudal point in the superior part of the caudal peduncle; 17: highest caudal point in the inferior part of the caudal peduncle; 18: end of the anal fin; 19: origin of the anal fin; 20: anal opening; 21: origin of the pelvic fin; 22: end of the pelvic fin radius; 23: nape, highest caudal point of the head.

**Figure 3 animals-11-01901-f003:**
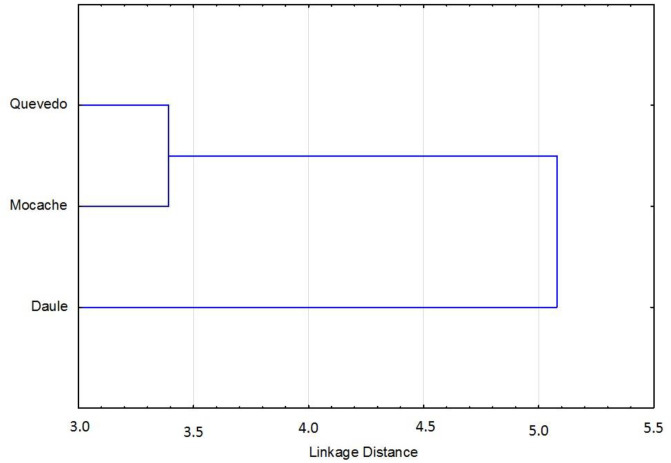
Clusters of Mahalanobis distances of *Brycon dentex* from three rivers.

**Table 1 animals-11-01901-t001:** Water characteristics in the three rivers of Guayas basin.

Indicators	Pintado River ^1^	Quevedo River ^2^	Mocache River ^3^
pH	7.72	8.23	7.11
Electric conductivity (μS/cm)	161	95.3	265.54
Temperature (°C)	25.6	19.2	26.11
Turbidity (NTU)	12.9	5.4	6.4
Total dissolved solids (mg/L)	110	96	133.5
Dissolved oxygen (OD, mg/L)	4.02	7.05	2.12

^1^ Gobierno Provincial de Manabí [47]. ^2^ Robin [48]. ^3^ Loor Castillo [49].

**Table 2 animals-11-01901-t002:** Morphometric measurements and meristic counts used to assess *Brycon dentex* in this study.

Measurement	Description	Acronym
Weight	Total weight including the gut and gonads	BW
Total length 1	Tip of the upper jaw to the top of the caudal superior end of the caudal fin	TL 1
Total length 2	Tip of the upper jaw to the top of the caudal inferior end of the caudal fin	TL 2
Standard length	Tip of the upper jaw to the tail base	SL
Head length	From the front of the upper lip to the posterior end of the opercula membrane	HL
Eye diameter	The greatest bony diameter of the orbit	ED
Pre-orbital length	Front of the upper lip to the cranial eye edge	Pre-OL
Pre-dorsal fin length	Front of the upper lip to the origin of the dorsal fin	Pre-DL
Pre-pectoral fin length	Front of the upper lip to the origin of the pectoral fin	Pre-PcL
Pre-pelvic fin length	Front of the upper lip to the origin of the pelvic fin	Pre-PvL
Pre-anal fin length	Front of the upper lip to the origin of the anal fin	Pre-AL
Dorsal fin length	From the base of the first dorsal spine to the base of the last dorsal ray	DFL
Dorsal fin ray length	From the base to the tip of the fifth dorsal ray	DFRL
Pectoral fin length	From the base to the tip of the pectoral fin	PcFL
Pelvic fin length	From the base to the tip of the pelvic fin	PvFL
Anal fin length	From the base of the first anal spine to the base of the last anal ray	AFL
Anal fin ray length	From the base to the tip of the last anal ray	AFRL
Upper jaw length	Straight line measurement between the snout tip and posterior edge of maxilla	UJL
Body perimeter 1	Body perimeter at the level of the first ray of the dorsal fin	P1
Body perimeter 2	Body perimeter at the level of the first radius of the anal fin	P2
Body perimeter 3	Body perimeter at the level of the last ray of the dorsal fin	P3
Body width 1	Straight line measurement from side to side at the level of the base of the first dorsal spine	LC1
Body width 2	Straight line measurement from side to side at the level of the base of the first anal spine	LC2
Body width 3	Straight line measurement from side to side at the level of the base of the last dorsal ray	LC3
Body depth 1	Body depth at the level of the first ray of the dorsal fin	AC1
Body depth 2	Body depth at the level of the first ray of the anal fin	AC2
Body depth 3	Body depth at the level of the first radius of the caudal fin	AC3
Dorsal fin rays	Number of thorns in the dorsal fin	DFR DFR
Pectoral fin rays	Number of thorns in the pectoral fin	PcFR
Pelvic fin rays	Number of thorns in the pelvic fin	PvFR
Anal fin rays	Number of thorns in the anal fin	AFR
Caudal fin rays	Number of thorns in the caudal fin	CFR

**Table 3 animals-11-01901-t003:** Descriptive statistics for body measurements (original data) in three populations of *Brycon dentex* (Mean ± SE (CV, %)).

Parameter ^1^	All	Pintado River (A)	Quevedo River (B)	Mocache River (C)	*p*-Value	
					A × C	B × C
BW	154.47 ± 5.78 (52.79)	172.61 ± 10.30 (41.78)	137.95 ± 8.43 (58.92)	165.82 ± 11.34 (51.61)	0.663	0.048
K	4.32 ± 0.17 (56.82)	5.10 ± 0.35 (48.50)	3.53 ± 0.22 (60.81)	4.93 ± 0.34 (51.78)	0.744	0.000
TL	21.46 ± 0.23 (15.41)	20.10 ± 0.46 (16.07)	22.52 ± 0.30 (13.02)	20.91 ± 0.45 (16.42)	0.217	0.003
SL	15.65 ± 0.16 (14.52)	15.20 ± 0.3 (13.62)	16.17 ± 0.22 (13.27)	15.2 ± 0.33 (16.35)	0.997	0.012
HL	5.65 ± 0.06 (16.03)	5.38 ± 0.11 (14.85)	5.90 ± 0.09 (15.40)	5.47 ± 0.12 (16.34)	0.605	0.005
ED	1.29 ± 0.02 (22.14)	1.27 ± 0.05 (25.59)	1.32 ± 0.03 (21.34)	1.27 ± 0.03 (20.35)	0.942	0.286
Pre-OL	1.19 ± 0.02 (19.94)	1.15 ± 0.04 (21.28)	1.23 ± 0.02 (17.57)	1.14 ± 0.03 (21.96)	0.801	0.020
Pre-DL	12.79 ± 0.14 (15.11)	12.28 ± 0.22 (12.45)	13.37 ± 0.20 (14.15)	12.29 ± 0.27 (16.81)	0.974	0.001
Pre-PcL	5.87 ± 0.08 (18.06)	5.76 ± 0.2 (24.42)	6.05 ± 0.09 (14.42)	5.68 ± 0.13 (16.98)	0.711	0.017
Pre-PvL	11.21 ± 0.12 (15.41)	11.02 ± 0.24 (15.11)	10.87 ± 0.16 (14.19)	11.93 ± 0.25 (15.72)	0.010	0.000
Pre-AL	15.39 ± 0.16 (14.53)	15.2 ± 0.32 (14.57)	14.90 ± 0.22 (14.11)	16.34 ± 0.29 (13.46)	0.009	0.000
DFL	2.54 ± 0.04 (22.18)	2.55 ± 0.11 (30.7.)	2.59 ± 0.05 (18.93)	2.45 ± 0.06 (17.69)	0.389	0.076
DFRL	3.52 ± 0.05 (19.13)	3.41 ± 0.07 (14.29)	3.47 ± 0.07 (20.65)	3.69 ± 0.09 (19.39)	0.024	0.075
PcFL	4.35 ± 0.05 (17.25)	4.20 ± 0.10 (16.1.)	4.47 ± 0.07 (16.09)	4.26 ± 0.11 (19.52)	0.703	0.096
PvFL	2.94 ± 0.04 (18.16)	2.79 ± 0.07 (16.86)	2.98 ± 0.05 (16.30)	3.00 ± 0.08 (21.21)	0.065	0.845
AFL	5.65 ± 0.1 (24.30)	5.06 ± 0.15 (21.27)	5.46 ± 0.15 (26.45)	6.44 ± 0.15 (17.23)	0.000	0.000
AFRL	2.51 ± 0.04 (24.09)	2.47 ± 0.08 (23.34)	2.56 ± 0.07 (24.73)	2.49 ± 0.08 (23.79)	0.866	0.492
UJL	1.02 ± 0.02 (27.92)	0.88 ± 0.02 (19.7)	1.11 ± 0.03 (25.23)	1.00 ± 0.04 (32.25)	0.024	0.036
AC1	5.89 ± 0.05 (12.04)	5.71 ± 0.13 (15.48)	5.99 ± 0.06 (9.59)	5.88 ± 0.1 (12.27)	0.280	0.292
AC2	5.44 ± 0.05 (12.93)	5.26 ± 0.14 (18.1)	5.53 ± 0.05 (9.00)	5.43 ± 0.1 (13.29)	0.314	0.303
AC3	2.37 ± 0.11 (66.14)	3.32 ± 0.41 (87.26)	2.05 ± 0.04 (18.96)	2.06 ± 0.05 (19.55)	0.002	0.896
P1	13.52 ± 0.19 (19.94)	12.35 ± 0.32 (18.17)	13.59 ± 0.26 (18.74)	14.41 ± 0.39 (20.48)	0.000	0.073
P2	13.99 ± 0.67 (67.41)	15.77 ± 2.64 (117.09)	13.52 ± 0.25 (18.11)	13.25 ± 0.4 (22.52)	0.313	0.556
P3	5.79 ± 0.06 (15.57)	5.29 ± 0.09 (11.32)	6.06 ± 0.09 (14.70)	5.77 ± 0.13 (16.68)	0.003	0.067
LC1	2.60 ± 0.03 (17.70)	2.38 ± 0.07 (20.06)	2.64 ± 0.04 (15.39)	2.74 ± 0.06 (17.15)	0.000	0.162
LC2	2.67 ± 0.04 (21.5)	2.40 ± 0.08 (24.70)	2.76 ± 0.05 (16.74)	2.75 ± 0.09 (23.81)	0.005	0.931
LC3	2.43 ± 0.04 (22.24)	2.13 ± 0.08 (25.00)	2.48 ± 0.05 (18.17)	2.58 ± 0.08 (22.74)	0.000	0.247

^1^ BW = body weight; K = Fulton’s factor; TL = total length; SL = standard length; HL = head length; ED = eye diameter; Pre-OL = pre-orbital length; Pre-DL = pre-dorsal fin length; Pre-PcL = pre-pectoral fin length; Pre-PvL = pre-pelvic fin length; Pre-AL = pre-anal fin length; DFL = dorsal fin length; DFRL = dorsal fin ray length; PcFL = pectoral fin length; PvFL = pelvic fin length; AFL = anal fin length; AFRL = anal fin ray length; UJL = upper jaw length; AC1 = body depth 1; AC2 = body depth 2; AC3 = body depth 3; P1 = body perimeter 1; P2 = body perimeter 2; P3 = body perimeter 3; LC1 = body width 1; LC2 = body width 2; LC3 = body width 3.

**Table 4 animals-11-01901-t004:** Discriminant functions for truss measurements in three populations of *Brycon dentex*.

Parameter ^1^	Wilks’ Lambda	Partial Lambda	F-Remove	*p*-Level	Toler	1-Toler
AFL	0.52	0.87	14.00	0.000	0.62	0.38
AC3	0.49	0.91	8.92	0.000	0.85	0.15
TL	0.49	0.91	8.74	0.000	0.74	0.26
P3	0.50	0.90	9.68	0.000	0.77	0.23
UJL	0.48	0.93	6.55	0.002	0.89	0.11
Pre-PvL	0.48	0.94	5.81	0.004	0.82	0.18
AFRL	0.47	0.95	4.80	0.009	0.72	0.28
Pre-DL	0.47	0.96	4.14	0.017	0.51	0.49
ED	0.47	0.95	4.32	0.015	0.64	0.36
P1	0.46	0.97	2.51	0.084	0.80	0.20
P2	0.45	0.99	1.02	0.363	0.63	0.37
Pre-PcL	0.45	0.99	1.09	0.338	0.63	0.37
LC3	0.47	0.96	3.94	0.021	0.19	0.81
LC2	0.46	0.97	2.89	0.058	0.20	0.80

^1^ TL = total length; ED = eye diameter; Pre-DL = pre-dorsal fin length; Pre-PcL = pre-pectoral fin length; Pre-PvL = pre-pelvic fin length; AFL = anal fin length; AFRL = anal fin ray length; UJL = upper jaw length; AC3 = body depth 3; P1 = body perimeter 1; P2 = body perimeter 2; P3 = body perimeter 3; LC2 = body width 2; LC3 = body width 3.

## Data Availability

The data are available on request from the corresponding author.

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
