# Peer review of "Morphological Variations of Wild Populations of Brycon dentex (Characidae, Teleostei) in the Guayas Hydrographic Basin (Ecuador). The Impact of Fishing Policies and Environmental Conditions"

_animals, 2021, doi:10.3390/ani11071901_

Round 1
Reviewer 1 Report
Review
Paper title: Morphological variations of wild populations of Brycon dentex in the Guayas hydrographic Basin (Ecuador). The impact of fishing policies and environmental conditions.
Little is known about the morphological variability of Brycon dentex, a native fish in rivers of Ecuador. The authors measured fish specimens from 3 different sites and related the differences found between the populations with environmental conditions and fishing pressure. They found that both groups of factors can affect some morphological parameters in the species and concluded that this approach (discriminant analysis) is useful in the management of Brycon dentex populations in Ecuador. Also, scientific information from this region is published mainly in Spanish and not available for international readers. This paper will be available for a wide audience that makes it more even important.
All these reasons explain the relevance of the paper by Ana Gonzalez-Martinez and co-authors submitted to "Animals".
General scores.
The data presented by the authors are original and significant. The study is correctly designed and technically sounds. In general, the statistical analyses are performed with good technical standards. We authors conducted careful work which will attract the attention of a wide range of specialists focused on the biology of commercial fish and other ichthyologists, fishermen, and fishery managers.
I have reviewed the previous version of this paper and indicated several concerns that the authors should address. The authors revised the text according to my recommendations and, in general, fixed the concerns.
Specific comments.
L 2. I think that more information about the object of this study is required in the title. I suggest including the systematic data as follows: Brycon dentex (Characidae, Teleostei)
L 33. Change “Test-T” to “t-tests”
L 38. Change “cluster 1” to “Cluster 1”
L 57. Change “morphologic analysis” to “morphological analysis”
L 60. Change “restauration” to “restoration”
L 86. km2. “2” should be superscript.
L 146. Change “Pupusá river” to “the Pupusá river”
L 148. Change “Pintado River are show in Table 1, values similar” to “the Pintado River are shown in Table 1, these values are similar”
L 173. Change “Pintado river” to “the Pintado river”
L 174. Change “in Quevedo” to “in the Quevedo”
L 176. Please, define the abbreviation “CV” and avoid use it in this context because CV is used to define Coefficient of Variation and is already used in the text (L 276)
L 181. Change “Pintado river” to “the Pintado river”
L 181. Change “most” to “the most”
L 182. Change “In Quevedo” to “In the Quevedo”
L 183. Change “more” to “most”
L 250. Change “Pintado” to “the Pintado”
L 245. Change “Pintado” to “the Pintado”
L 257. Change “Quevedo” to “the Quevedo”
L 306. Change “capture fish” to “captured fish”
L 330-331. Change “rivers of Hypothesis 1, probably both fishing policies are closer than what it was initially considered” to “these rivers (Hypothesis 1), more likely because the local fishing regimes are closer than it was initially considered”
L 348. The role of introduced fish is still unclear. May be the authors would like to say that the fish invaders are competitors with native species for food and space and, therefore, native fish have less resources for somatic growth than at non-invaded sites. Please, clarify.
L 349. Change “Quevedo” to “the Quevedo”
L 358. Change “the Quevedo and the Mocache” to “the Quevedo and Mocache”
L 392. Change “Quevedo” to “the Quevedo”
L 421. Change “economic” to “low-cost”, “in situ” should be italicized
L 426. Change “specie” to “species”
L 436-437. Change “Both factors could be considered direct drivers of diversity maintenance.”
L 454. Change “Latinoamerica” to “Latin America”
Author Response
Review Report Form #1
RESPONSE
We would like to thank for your work and valuable comments that have substantially help us to improve the quality of our initial manuscript. We have considered your comments and we have done all the suggested corrections. To improve our work, we highlight the changes asked for you and rest of reviewers in bold letters. Also, we have sent to MPDI English Edition the manuscript to proofread.
We hope that you like the new version of the manuscript.
Thank you very much for your attention.
Kind Regards,
The authors
Open Review
(x) I would not like to sign my review report
( ) I would like to sign my review report
English language and style
( ) Extensive editing of English language and style required
(x) Moderate English changes required
( ) English language and style are fine/minor spell check required
( ) I don't feel qualified to judge about the English language and style
|
Yes |
Can be improved |
Must be improved |
Not applicable |
|
|
Does the introduction provide sufficient background and include all relevant references? |
(x) |
( ) |
( ) |
( ) |
|
Is the research design appropriate? |
(x) |
( ) |
( ) |
( ) |
|
Are the methods adequately described? |
(x) |
( ) |
( ) |
( ) |
|
Are the results clearly presented? |
(x) |
( ) |
( ) |
( ) |
|
Are the conclusions supported by the results? |
( ) |
(x) |
( ) |
( ) |
Comments and Suggestions for Authors
Review
Paper title: Morphological variations of wild populations of Brycon dentex in the Guayas hydrographic Basin (Ecuador). The impact of fishing policies and environmental conditions.
Little is known about the morphological variability of Brycon dentex, a native fish in rivers of Ecuador. The authors measured fish specimens from 3 different sites and related the differences found between the populations with environmental conditions and fishing pressure. They found that both groups of factors can affect some morphological parameters in the species and concluded that this approach (discriminant analysis) is useful in the management of Brycon dentex populations in Ecuador. Also, scientific information from this region is published mainly in Spanish and not available for international readers. This paper will be available for a wide audience that makes it more even important.
All these reasons explain the relevance of the paper by Ana Gonzalez-Martinez and co-authors submitted to "Animals".
General scores.
The data presented by the authors are original and significant. The study is correctly designed and technically sounds. In general, the statistical analyses are performed with good technical standards. We authors conducted careful work which will attract the attention of a wide range of specialists focused on the biology of commercial fish and other ichthyologists, fishermen, and fishery managers.
I have reviewed the previous version of this paper and indicated several concerns that the authors should address. The authors revised the text according to my recommendations and, in general, fixed the concerns.
Specific comments.
L 2. I think that more information about the object of this study is required in the title. I suggest including the systematic data as follows: Brycon dentex (Characidae, Teleostei)
Answer: The correction has been made
L 33. Change “Test-T” to “t-tests”
Answer: The correction has been made
L 38. Change “cluster 1” to “Cluster 1”
Answer: The correction has been made
L 57. Change “morphologic analysis” to “morphological analysis”
Answer: The correction has been made
L 60. Change “restauration” to “restoration”
Answer: The correction has been made
L 86. km2. “2” should be superscript.
Answer: The correction has been made
L 146. Change “Pupusá river” to “the Pupusá river”
Answer: The correction has been made
L 148. Change “Pintado River are show in Table 1, values similar” to “the Pintado River are shown in Table 1, these values are similar”
Answer: The correction has been made
L 173. Change “Pintado river” to “the Pintado river”
Answer: The correction has been made
L 174. Change “in Quevedo” to “in the Quevedo”
Answer: The correction has been made
L 176. Please, define the abbreviation “CV” and avoid use it in this context because CV is used to define Coefficient of Variation and is already used in the text (L 276)
Answer: We have clarified the abbreviation “CV” as “cv, measurement of power where 1 cavallo vapore (cv) = 0.9862 horsepower (hp)”
L 181. Change “Pintado river” to “the Pintado river”
Answer: The correction has been made
L 181. Change “most” to “the most”
Answer: The correction has been made
L 182. Change “In Quevedo” to “In the Quevedo”
Answer: The correction has been made
L 183. Change “more” to “most”
Answer: The correction has been made
L 250. Change “Pintado” to “the Pintado”
Answer: The correction has been made
L 245. Change “Pintado” to “the Pintado”
Answer: The correction has been made
L 257. Change “Quevedo” to “the Quevedo”
Answer: The correction has been made
L 306. Change “capture fish” to “captured fish”
Answer: The correction has been made
L 330-331. Change “rivers of Hypothesis 1, probably both fishing policies are closer than what it was initially considered” to “these rivers (Hypothesis 1), more likely because the local fishing regimes are closer than it was initially considered”
Answer: The correction has been made
L 348. The role of introduced fish is still unclear. May be the authors would like to say that the fish invaders are competitors with native species for food and space and, therefore, native fish have less resources for somatic growth than at non-invaded sites. Please, clarify.
Answer: As your indications, we have cleared this aspect adding the following sentence: “They are also competitors with native species for food and space, and, therefore, native fish have fewer resources for somatic growth than at non-invaded sites.”
L 349. Change “Quevedo” to “the Quevedo”
Answer: The correction has been made
L 358. Change “the Quevedo and the Mocache” to “the Quevedo and Mocache”
Answer: The correction has been made
L 392. Change “Quevedo” to “the Quevedo”
Answer: The correction has been made
L 421. Change “economic” to “low-cost”, “in situ” should be italicized
Answer: The correction has been made
L 426. Change “specie” to “species”
Answer: The correction has been made
L 436-437. Change “Both factors could be considered direct drivers of diversity maintenance.”
Answer: The correction has been made
L 454. Change “Latinoamerica” to “Latin America”
Answer: The correction has been made

Reviewer 2 Report
Dear Authors,
I very much appreciated your care in responding to the comments of the first submission, and the answers regarding the genetic support and the resonance of the work, now well clarified and reported in various parts of the text. The manuscript is now much improved in its various parts and written with more care than the previous version, and in this way its general quality has increased and is in my opinion considerable for publication. What I ask is that you rely on an English language editing service to improve your writing substantially and make it more effective for the reader.
Best regards
The reviewer
Author Response
Review Report Form #2
RESPONSE
We would like to thank for your work and valuable comments that have substantially help us to improve the quality of our initial manuscript. We have considered your comments and we have done all the suggested corrections. To improve our work, we highlight the changes asked for you and rest of reviewers in bold letters. Also, we have sent to MPDI English Edition the manuscript to proofread.
We hope that you like the new version of the manuscript.
Thank you very much for your attention.
Kind Regards,
The authors
Open Review
( ) I would not like to sign my review report
(x) I would like to sign my review report
English language and style
(x) Extensive editing of English language and style required
( ) Moderate English changes required
( ) English language and style are fine/minor spell check required
( ) I don't feel qualified to judge about the English language and style
|
Yes |
Can be improved |
Must be improved |
Not applicable |
|
|
Does the introduction provide sufficient background and include all relevant references? |
(x) |
( ) |
( ) |
( ) |
|
Is the research design appropriate? |
(x) |
( ) |
( ) |
( ) |
|
Are the methods adequately described? |
(x) |
( ) |
( ) |
( ) |
|
Are the results clearly presented? |
( ) |
(x) |
( ) |
( ) |
|
Are the conclusions supported by the results? |
(x) |
( ) |
( ) |
( ) |
Comments and Suggestions for Authors
Dear Authors,
I very much appreciated your care in responding to the comments of the first submission, and the answers regarding the genetic support and the resonance of the work, now well clarified and reported in various parts of the text. The manuscript is now much improved in its various parts and written with more care than the previous version, and in this way its general quality has increased and is in my opinion considerable for publication. What I ask is that you rely on an English language editing service to improve your writing substantially and make it more effective for the reader.
Answer: The manuscript has been sent to MPDI English Edition to proofread.

Reviewer 3 Report
Despite have provided comments and suggestions in two rounds of review, this study still does not explain how body shape is meant to reflect fishing pressures or environmental conditions. The intro provides a poor theoretical connection linking environmental factors and body shape. The main claim of the summary is poorly supported: “The status of conservation of a native fish species is often a key indicator of the state of alteration of habitats.” This connecton always depends on the species and the habitats. The study does not explain or justify the choice of Brycon dentex as a study organism. Despite comments from two previous rounds of review this study still reads like results of field work looking for a question to answer, and yet not finding it.
Author Response
Review Report Form #3
RESPONSE
We would like to thank for your work and valuable comments that have substantially help us to improve the quality of our initial manuscript. We have considered your comments and we have done all the suggested corrections. To improve our work, we highlight the changes asked for you and rest of reviewers in bold letters. Also, we have sent to MPDI English Edition the manuscript to proofread.
We hope that you like the new version of the manuscript.
Thank you very much for your attention.
Kind Regards,
The authors
Comments and Suggestions for Authors
1/ Despite have provided comments and suggestions in two rounds of review, this study still does not explain how body shape is meant to reflect fishing pressures or environmental conditions. The intro provides a poor theoretical connection linking environmental factors and body shape. The main claim of the summary is poorly supported: “The status of conservation of a native fish species is often a key indicator of the state of alteration of habitats.” This connection always depends on the species and the habitats. The study does not explain or justify the choice of Brycon dentex as a study organism. Despite comments from two previous rounds of review this study still reads like results of field work looking for a question to answer, and yet not finding it.
Answer: The first paragraph of introduction has been changed, being the new the following one:
“The ecological theory of diversification [1] and studies of wild populations explain how changes in environmental factors could induce changes in behavior, morphology, and physiology [2,3]. Selection pressure in new environments favors the divergence of populations, and there is a strong link between environmental variations and the morphological diversification of a population [2,4,5]. Furthermore, habitat modification may result in changes in the composition, geographical spread, and population structure of a species [6-8], while diversity is considered an indicator of ecosystem restoration [9]. In this sense, fishing policies in each country seek food sovereignty and the maintenance of genetic resources and biodiversity through economic incentives and regulatory measures [10].”
Also, we have rewritten the fourth paragraph to justify the choice of Brycon dentex as following:
Brycon dentex Günther 1860 (pez dama), from the Bryconidae family, is a native species that is widely distributed in western Ecuador in different rivers of the CHG [31-33], in the Tahuín dam near Peru [34], and in the North of Perú [35]. In a previous study, our research group conducted a preliminary morphology characterization of Brycon dentex [36]. Morphologically, this species has a single dorsal fin and two bifurcations in the caudal fin, a total length of 51 cm, and a powerful upper jaw. It is considered an omnivorous species [33,37]. Sampling in natural environments has shown that it can reach sexual maturity at lengths of 20-26 cm [37]. Its conservation status is of “least concern”, and it is included in the IUCN Red List.
The Brycon dentex, the subject of our study, is widely geographical spread across a broad range of ecological conditions accompanied by equally diverse morphological variations in Ecuador. It is not included on the IUCN Red List, although there is insufficient data for characterization. Brycon dentex has an omnivorous mode of feeding and a strong ability to adapt rapidly to different environmental conditions. In addition to characterizing Brycon dentex, it is of great interest to relate its morphological variability to biodiversity maintenance. The variation among the stocks of river populations could be a consequence of phenotypic plasticity in response to unusual hydrological conditions [2,17].
Ferrito et al. [38] and Mir et al. [39] have conducted similar studies on other freshwater species, and Dasgupta et al. [40] stated that morphological discrimination in various populations are strongly influenced by habitat differences. Growth variations also occur in response to different habitats [41]. Under the hypothesis that the morphological differences between populations of native freshwater species could be used as bioindicators, the causes of these differences were classified into anthropogenic and habitat modifications [42]. Knowledge in this area can be used as a tool for both smart and ecological management of resources [43,44]. In addition, there is a lack of knowledge about the morphological characterization of Brycon dentex and the variation in the traits of different rivers in the Guayas basin. In this context, this study aimed to contribute to the attainment of an adequate management ecosystem equilibrium and the characterization of animal genetic resources.
2/ This study aims to document variation in body shape among three populations of a commercially harvested fish species from the Guayas river basin in Ecuador, and to correlate aspects of body shape variation with particular management practices and environmental conditions. The study design is basically sound and the results may indeed match these aims, but these results are not clearly presented. Rather than using a distance (Mahalanobis) metric (Fig. 3) to report shape differences it is more useful and common these days to use a multivariate statistical analyses (e.g. PCA, DCA or similar), which provide loadings of individual traits. The main points of the statistical analysis buried in Table 2 and 3 should be presented graphically, and the trait variables demonstrated to vary significantly with management and environmental conditions could be plotted on maps. Finally, the English language is awkward and ambiguous in many places, and the whole document must be carefully edited.
Answer: It is correct what the reviewer indicates, although we would like to clarify the methodology developed. The final result is similar, although the paths to reach it are differentiated.
Option a: When data on a population are available and subpopulations are to be differentiated, PCA, Multiple Correspondence, PC, etc., are usually used and later, Cluster. That is, in a first phase the dimensionality (number of variables) is reduced and in a second, factors (latent variables) are built. Subsequently, a rotation of the factors is made until the groups or subpopulations are generated (homogeneous within themselves and heterogeneous among them). In this case we do not know a priori how they will be grouped. Main references are:
- Stylianou, A., Sdrali, D., & Apostolopoulos, C. D. (2020). Capturing the diversity of Mediterranean farming systems prior to their sustainability assessment: The case of Cyprus. Land Use Policy, 96, 104722.
- Rangel Quintos, J., Perea, J., Pablos-Heredero, C. D., Espinosa García, J. A., Toro Mújica, P. M., Feijoo, M., & García Martínez, A. R. (2020). Structural and Technological Characterization of Tropical Smallholder Farms of Dual-Purpose Cattle in Mexico. Animals 2020, 10, 86
- Rivas, J., Perea, J., Angón, E., Barba, C., Morantes, M., Dios-Palomares, R., & García, A. (2015). Diversity in the dry land mixed system and viability of dairy sheep farming. Italian Journal of Animal Science, 14(2), 3513.
Option b: When a fix factor is used a priori to differentiate subpopulations. Multivariate methods are usually applied, combining cluster with discriminant function, which allows identifying those variables responsible for these differences and with high classification power. Main references are:
- Gonzalez-Martinez, A.: De-Pablos-Heredero, C.; González, M.; Rodriguez, J.; Barba, C.; García, A. Usefulness of discriminant analysis in the morphometric differentiation of six native freshwater species from Ecuador. Animals, 2021, 11 (1), 111. https://doi.org/10.3390/ani11010111.
- Gonzalez-Martinez, A.; Lopez, M.; Molero, H.M.; Rodriguez, J.; González, M.; Barba, C.; García, A. Morphometric and Meristic Characterization of Native Chame Fish (Dormitator latifrons) in Ecuador Using Multivariate Analysis. Animals 2020, 10, 1805; https://doi.org/10.3390/ani10101805.
- Caez, J.; González, A.; González, M.A.; Angón, E.; Rodríguez, J.M.; Peña, F.; Barca, C.; García, A. Application of multifactorial discriminant analysis in the morphostructural differentiation of wild and cultured populations of Vieja Azul (Andinoacara rivulatus). J. Zool. 2019, 43, 516–530.
In our research the aim was to investigate whether three wild populations of Brycon dentex in the Guayas Hydrographic Basin (CHG) have undergone significant morphological diversifications. So, we think that it is more appropriate to use the Discriminant analysis to establish differences among the three groups (hypothesis 3).

Reviewer 4 Report
The article is very interesting and deals with an important issue of „Morphological variations of wild populations of Brycon dentex in the Guayas hydrographic Basin (Ecuador). The impact of fishing policies and environmental conditions „, there are errors in the article that need to be corrected and certain deficiencies that need to be addressed.
My comments:
Introduction
The names of fish species, when mentioned for the first time in the text, should include the Latin name, the author and the date, e.g. Brycon dentex Günther 1860– lines 88.
What does the term "white water" mean? – line 136
Material and methods
Please improve the quality of the Figure 1 and 2. Figure 1 - please enlarge numbers.
Please correct table 3:
- Body width 1; Body width 2; Body width 3 - have the same description?
Results
Lines 215 and 262 – there are two Tables 2. Line 262 - should be Table 3!!!!
Line 243, 248, 255 - should by Table 3
Line 292 – should by Table 4 ! Line 284 - should by Table 4! Please correct the numbering of the tables throughout the text !!!
References
Line 178 Pacheco- Bedoya [45] – references is [47]?
Make a correction of the numbering of the References!
Author Response
Review Report Form #4
RESPONSE
We would like to thank for your work and valuable comments that have substantially help us to improve the quality of our initial manuscript. We have considered your comments and we have done all the suggested corrections. To improve our work, we highlight the changes asked for you and rest of reviewers in bold letters. Also, we have sent to MPDI English Edition the manuscript to proofread.
We hope that you like the new version of the manuscript.
Thank you very much for your attention.
Kind Regards,
The authors
Open Review
(x) I would not like to sign my review report
( ) I would like to sign my review report
English language and style
( ) Extensive editing of English language and style required
( ) Moderate English changes required
( ) English language and style are fine/minor spell check required
(x) I don't feel qualified to judge about the English language and style
|
Yes |
Can be improved |
Must be improved |
Not applicable |
|
|
Does the introduction provide sufficient background and include all relevant references? |
(x) |
( ) |
( ) |
( ) |
|
Is the research design appropriate? |
(x) |
( ) |
( ) |
( ) |
|
Are the methods adequately described? |
( ) |
(x) |
( ) |
( ) |
|
Are the results clearly presented? |
( ) |
(x) |
( ) |
( ) |
|
Are the conclusions supported by the results? |
( ) |
(x) |
( ) |
( ) |
Comments and Suggestions for Authors
The article is very interesting and deals with an important issue of „Morphological variations of wild populations of Brycon dentex in the Guayas hydrographic Basin (Ecuador). The impact of fishing policies and environmental conditions „, there are errors in the article that need to be corrected and certain deficiencies that need to be addressed.
My comments:
Introduction
The names of fish species, when mentioned for the first time in the text, should include the Latin name, the author and the date, e.g. Brycon dentex Günther 1860– lines 88.
Answer: the correction has been made
What does the term "white water" mean? – line 136
Answer: We have added “or with high oxygen concentration” to clear this point
Material and methods
Please improve the quality of the Figure 1 and 2. Figure 1 - please enlarge numbers.
Answer: We have improved the quality of both figures. To improve the quality of Figure 1, we have changed the imagen of CHG. To enlarge numbers, we have changed the specimen used in the figure 2,
Please correct table 3:
- Body width 1; Body width 2; Body width 3 - have the same description?
Answer: It is a measure of body development, but at different levels; AC1 measures the depth of the body at the height of the first ray of the dorsal fin; AC2 measures it at the height of the first ray of the anal fin; and AC3 measures it at the height of the first radius of the tail fin
Results
Lines 215 and 262 – there are two Tables 2. Line 262 - should be Table 3!!!!
Answer: the correction has been made
Line 243, 248, 255 - should by Table 3
Answer: the correction has been made
Line 292 – should by Table 4 ! Line 284 - should by Table 4! Please correct the numbering of the tables throughout the text !!!
Answer: the correction has been made
References
Line 178 Pacheco- Bedoya [45] – references is [47]? Make a correction of the numbering of the References!
Answer: the correction has been made

This manuscript is a resubmission of an earlier submission. The following is a list of the peer review reports and author responses from that submission.
Round 1
Reviewer 1 Report
Dear Authors,
I found your manuscript of low quality and resonance, very limited to the studied area which, however important it may be at the regional level, without further analysis is likely to be very limited to the level of interest for the readers; indeed, as you have stated in discussion section:
" The current characterization carried out is of a preliminary nature since it only considers the morphological aspects. These interactions have not been considered in this research. To determine the relative contribution of genetic and plasticity (i.e. environmentally induced phenotypic differences), genetic analyses and common garden rearing experiments would be a necessary next step. "
Nowadays the genetic support on these kind of study is almost essential, to give more resonance to the related manuscripts, for this reason my opinion on the quality of your manuscript is low.
Moreover, before considering a publication in Animals Journal, a professional extensive English editing it's necessary, to give it more clarity and fluency, that are essential.
The lines between 132-151 probably should be included in introduction section, as aim and scope statement support.
Please be sure in the main text to contextualized you data and refer to average data when you write about morphological values measured, there are several of these mistakes disseminated in the manuscript.
In materials and methods section no mention has been made of the trapping patterns of animals which may be relevant to the collection of data in this kind of studies.
Please take care of all these suggestions before a resubmission in Animals Journal.
Best regards
The reviewer
Reviewer 2 Report
Review
Paper title: Morphological variations of wild populations of Brycon dentex in the Guayas hydrographic Basin (Ecuador). The impact of fishing policies and environmental conditions.
Little is known about the morphological variability of Brycon dentex, a native fish in rivers of Ecuador. The authors measured fish specimens from 3 different sites and attired to relate the differences found between the populations with environmental conditions and fishing pressure. They found that both groups of factors can affect some morphological parameters in the species and concluded that this approach (discriminant analysis) is useful in the management of Brycon dentex populations in Ecuador.
All these reasons explain the relevance of the paper by Ana Gonzalez-Martinez and co-authors submitted to "Animals".
General scores.
The data presented by the authors are original and significant. Some conclusions should be explained more precisely. The study is correctly designed and technically sounds. In general, the statistical analyses are performed with good technical standards. We authors conducted careful work which will attract the attention of a wide range of specialists focused on the biology of commercial fish and other ichthyologists, fishermen, and fishery managers.
Major concerns.
L 108-114. For this river, the authors should provide the same descriptive information as they did for Mochache and Quevedo rivers (flow velocity, oxygenation level, clean and waste water).
L 191. Test-T? Did the authors use the Student t-test? If so, they should add information about the normality and heteroscedasticity of their data.
L 260. The authors should explain why the fishes from their sites were smaller than reported by Revelo [47] and Revelo and Laaz [29].
L 285. The authors should explain more clearly how different fishing regimes (fishing gears?) can affect body depth, perimeter and width in Bryncon dentex. A traditional opinion is that high fishing pressure leads to a decrease in body length and size at maturity. In the case of Bryncon dentex, differences in BW, K, TL, and SL between Pintado and Mochache rivers were insignificant (Table 2). Please, clarify.
L 288. The authors declared that “introduction of non-native species” can affect morphometric parameters in Bryncon dentex. The role of non-indigenous species was not tested in this study, therefore, the authors should explain this statement. The references they used here are not available for a wide audience.
L 326-328. The authors should explain these associations more clearly. Which mechanisms exist in each case (AFL, UJL, Pre-Pvl, AC3, and P3)?
Conclusion. The authors should indicate key factors affecting the morphological parameters of Brycon dentex according to their statements in the “Introduction” section (L. 100–101).
Specific comments.
L 21. Change “proposal” to “a proposal”
L 24. Change “and thus be” to “and, thus,”
L 27. Change “Most part” to “Most of them”
L 31. Change “measurements” to “parameters were measured”
L 35. Change “associated to” to “associated with”
L 36. Change “condition” to “conditions”
L 40. Change “as indicator” to “as an indicator”
L 44. Change “highlight” to “explain”
L 47. Change “the biodiversity” to “biodiversity”
L 49. Change “structure of the species” to “structure of species”
L 49-50. Change “diversity of species” to “diversity”
L 51. Change “shown by a native freshwater specie” to “in a native freshwater species”
L 54. Change “life history traits” to “life-history traits”
L 55. Change “management program” to “management programs”
L 56. Change “ecological conditions” to “environmental conditions”
L 60. Change “diets composition” to “diet composition”
L 62. Change “differentiation of populations” to “the differentiation of populations”
L 66. Change “associated to diverse” to “associated with diverse”
L 69. Change “in classification” to “in the classification”
L 70. Change “most biodiversity of fish species” to “to most fish species”
L 71-72. Change “This is the case of Ecuador,” to “Ecuador ”
L 77. Change “previous analysis” to “a previous study”
L 83-84. Change “morphological variability of a freshwater specie with biodiversity maintenance” to “their morphological variability to biodiversity maintenance”
L 88-89. Change “Morphological and growth variations were responses adapted to different habitats [33].” to “Growth variations are also responses to different habitats [33].”
L 91. Change “anthropological” to “anthropogenic”
L 97. Change “the aim of research was” to “the aim of our research was”
L 101. Change “identify key factors” to “identify of key factors”
L 102. Change “development of diversity conservation programs as sustainable fishing” to “the development of diversity conservation programs and sustainable fishing”
L 106. Change “simple size” to “sample size” (twice)
L 107. Change “simple size” to “sample size”
L 115. Change “show white” to “shows white”
L 117. Change “slowly water, poorly oxygen and high level” to “slow water, poor oxygen and a high level”
L 118. Change “province of Manabi” to “the province of Manabi”
L 125. Change “Physic characteristics water” to “Water characteristics”
L 131. Change “sampling size” to “sample size”
L 133. Change “phenotypic” to “the phenotypic”
L 149. Change “slowly water” to “low flow velocity”
L 151. Change “slowly water” to “low flow velocity”
L 152-154. Change “The specimens of Brycon dentex were caught between January and March 2019 by the fishermen, with a size greater than 56 g and 12.38 cm, and phenotypic characteristics of the species.” to “The specimens of Brycon dentex (weight > 56 g, body length > 12.38 mm) were caught by the fishermen between January and March 2019”
L 159. Change “A same person” to “The same person”
L 180. Change “indicator of animal welfare and widely used” to “a widely-used indicator of animal welfare”
L 188. Change “Bartlett's test” to “Bartlett's tests”
L 189. Change “arithmetic mean” to “the arithmetic mean”
L 193. Change “the hypothesis” to “hypothesis”
L 202. " Brycon dentex" should be italicized.
L 204. Change “medium level of homogeneity with low coefficient of variation” to “a medium level of homogeneity with low coefficients of variation”
L 205. Change “significative” to “significant”
L 206. Change “no presented” to “not presented”
L 209. Change “the morphometric comparison results” to “the results of morphometric comparisons”
L 210. Change “with fishing management differentiated” to “with different fishing management”
L 211. Change “was shown” to “are shown”
L 212-213. Change “In 11 over 25 morphometric measures were found significative (P < 0.05) as differences between both rivers” to “Eleven from the 25 morphometric measures showed significant differences (P < 0.05) between Pintado and Mocache rivers”
L 214. Change “Regarding to hypothesis” to “Regarding hypothesis”
L 216-219. Change “In 10 over 25 variables significative differences between fish populations in Quevedo and Mocache rivers were found (P < 0.05), being TL, SL, HL, Pre-OL, Pre-DL, Pre-PcL, Pre-PvL, Pre-AL, AFL and UJL.” to “Significant differences between Quevedo and Mocache rivers were found for 10 morphometric variables including TL, SL, HL, Pre-OL, Pre-DL, Pre-PcL, Pre-PvL, Pre-AL, AFL and UJL.”
Table 2, Column 1. Change “Character” to “Parameter”
L 230. Please check “** p < 0.001; *** p < 0.001”. There are the same values for “**” and “***”. May be “** p < 0.01; *** p < 0.001”? Similar concern is for L 254.
L 232. Change “on number” to “on the number”
L 233. Change “y 19.01, respectively (data no presented)” to “and 19.01, respectively”
L 234. Change “The variations thought coefficient of variation were low” to “Coefficients of variation were low”
L 235. Change “values the three populations” to “values for the three populations”, “with percentage” to “with a percentage”
L 238. Change “significative” to “significant”
L 239. Change “most discriminant variables” to “major discriminant variables”
L 242. Change “different morphology model” to “specific morphology model”
Table 3, Column 1. Change “Character” to “Parameter”
L 259. The authors stated “21.56 cm total length” whereas in Table 1, TL = 21.46 cm. Please, check.
L 259. Change “face” to “compared”
L 260. Change “23.7 y 23.3” to “23.7 and 23.3”.
L 269. Change “of habitat” to “of the habitat”
L 272. Delete “, among others”, Change “In Quevedo” to “In the Quevedo”
L 278, 281, 282. Change “Pintado” to “the Pintado”
L 280. Change “width body” to “width of body”
L 281. Change “more” to “greater”
L 282. Change “more” to “greater”
L 283. Change “as reservoir” to “as a reservoir”
L 285. Change “captures frequency and size” to “both frequency and size of captured fish”
L 291. Change “was” to “were”
L 293. Change “Quevedo river” to “the Quevedo river”
L 294. Change “Quevedo river” to “the Quevedo river”
L 297. Change “Quevedo” to “the Quevedo”
L 298. Change “Mocache “ to “the Mocache “, “was slower” to “is slower”
L 300-302. Change “Nevertheless, the most relevant factor condition in Mocache river could be in the fact that this species” to “Food availability seems to be the most relevant factor in the Mocache river because this fish”
L 308-309. Change “cooperatives, fishing off the fishing season.” to “cooperatives and fishing efforts during the fishery season.”
L 316. Change “discriminant model” to “the discriminant model”
L 331. Change “introduction” to “the introduction”
L 336. Change “a situation of danger of extinction” to “dangerous situations in terms of fish extinction”, “Anthropomorphic action” to “Anthropogenic activities”
L 337. Change “the land use” to “land use”
L 341. Change “resources conservation” to “resource conservation”
L 350-351. Delete “common garden”
L 353-355. Delete.
L 358. Change “associated” to “were associated”
L 382-383. Delete “for research and the code of practices for the housing and care of animals used in scientific procedures”.
L 384-386. Change “This is not applicable as the data are not in any data repository of public access, however if editorial committee needs access, we will happily provide them, please use this email: pa1gamaa@uco.es.” to “The data are available on request from the corresponding author.”